# Comparing Protection of Remote Limb with Resveratrol Preconditioning following Rodent Subarachnoid Hemorrhage

**DOI:** 10.3390/biom12040568

**Published:** 2022-04-12

**Authors:** Sebastian Koch, Giselle De La Rua, Donnae Farquharson, Isabel Saul, Miguel Perez-Pinzon, Kunjan Dave

**Affiliations:** 1Department of Neurology, Miller School of Medicine, University of Miami, Miami, FL 33136, USA; gmd106@miami.edu (G.D.L.R.); dxf416@miami.edu (D.F.); isaul@med.miami.edu (I.S.); perezpinzon@med.miami.edu (M.P.-P.); kdave@med.miami.edu (K.D.); 2Peritz Scheinberg Cerebral Vascular Disease Research Laboratories, Miller School of Medicine, University of Miami, Miami, FL 33136, USA; 3Neuroscience Program, Miller School of Medicine, University of Miami, Miami, FL 33136, USA

**Keywords:** preconditioning, subarachnoid hemorrhage, resveratrol, limb preconditioning

## Abstract

Background: Preventing delayed cerebral ischemia (DCI) after subarachnoid hemorrhage (SAH) remains an important therapeutic target. Preconditioning stimulates multiple endogenous protective mechanisms and may be a suitable treatment for DCI following SAH. We here compare remote limb conditioning with resveratrol conditioning in a clinically relevant SAH model. Methods: We produced a SAH in 39 male Sprague Dawley rats using a single injection model. Animals were randomized to four groups: repetitive limb conditioning with a blood pressure cuff, sham conditioning, intraperitoneal resveratrol (10 mg/kg) or intraperitoneal vehicle administered at 24, 48 and 72 h after SAH. On day 4 neurological and behavioral scores were obtained, and animals were euthanized. The cross-sectional area of the basilar artery was measured at the vertebrobasilar junction, and at the mid and distal segments. Hippocampal cells were counted in both hemispheres and normalized per mm length. We compared true limb preconditioning with sham conditioning and resveratrol with vehicle preconditioning. Results: The cross-sectional area of the mid-basilar artery in the true limb preconditioning group was significantly larger by 43% (*p* = 0.03) when compared with the sham preconditioning group. No differences in the cross-sectional area were found in the resveratrol-treated group when compared to the vehicle-treated group. We found no differences in the neuro score, behavioral score, and in mean hippocampal neuron counts between the groups. Conclusion: We found beneficial vascular effects of remote limb preconditioning on SAH-induced basilar artery vasoconstriction. Our findings support further studies of limb preconditioning as a potential treatment after SAH.

## 1. Introduction

Cerebral vasoconstriction complicates the management of human subarachnoid hemorrhage (SAH) in 20–30% of cases and contributes to delayed cerebral ischemia (DCI) [1]. DCI remains a significant, but potentially preventable morbidity of SAH. DCI predictably occurs within 4–14 days of SAH, which allows implementation of a potentially protective intervention. Preconditioning may be such an intervention. Preconditioning describes the phenomenon that sublethal insults protect organisms from later more severe insults, by stimulating endogenous defense mechanisms.

SAH is an ideal clinical setting for translational studies of preconditioning [2]. The basic premise is that conditioning is implemented prior to cerebral ischemia. This may be impractical in spontaneous ischemic stroke. However, in SAH the risk period for DCI is well known and preconditioning can be readily instituted in anticipation of cerebral ischemia. To be clinically relevant, experimental SAH models must meet several translational needs. The conditioning stimulus must be practically implementable in humans. There should be a delay between SAH and starting the conditioning to simulate the delay in evaluation and treatment of the ruptured aneurysm in the clinical setting. Conditioning needs to be repetitive and continue over 14 days, which is the high-risk period for DCI in humans. 

The optimal stimulus for preconditioning is not known. Many different stimuli have been shown to be protective. This includes pharmacological conditioning, temperature, metabolic (hypoglycemia), and conditioning remotely with ischemia, such as limb preconditioning. We have previously shown that resveratrol, a preconditioning agent, mimics ischemic conditioning in its protective effect and mechanism of action through Sirtuin activation [3,4]. However, there have been very few studies directly comparing different conditioning strategies. In the present study, our objective was to investigate the protective effects of two preconditioning stimuli in a clinically relevant rodent model of SAH. We compared limb preconditioning with resveratrol preconditioning. We used a research design and treatment paradigm to simulate human SAH as much as possible, i.e., SAH was first induced, the conditioning intervention was started after 24 h, and was repeated over three days. 

## 2. Methods

Animal studies were conducted as per the Guide for the Care and Use of Laboratory Animals published by the National Institutes of Health and protocols approved by the Animal Care and Use Committee at the University of Miami on 16 May 2017 protocol #17-079. 

### 2.1. Induction of SAH

We used a single injection experimental SAH model in 39 young (8–12 weeks old) male Sprague Dawley rats (Charles River Laboratories, Wilmington, MA, USA) as described previously. [5] Prior to surgery, animals were randomly assigned to four different groups: (1) sham remote limb preconditioning-treated group, (2) remote limb conditioning-treated group, (3) intraperitoneal vehicle-treated (0.9% saline containing 6.66% dimethyl sulfoxide) group, and (4) intraperitoneal resveratrol (10 mg/kg)-treated group. All surgeries were done blinded to group allocation.

Anesthesia was induced with isoflurane in a mixture of 70% nitrous oxide and 30% oxygen. Temperature probes were inserted into the rectum and the left temporalis muscle, and separate heating lamps were used to maintain rectal and cranial temperatures at 37 °C to 37.5 °C throughout the surgery. A polyethylene catheter was introduced into the right femoral artery for blood collection.

Rats were turned prone on the table and shaven over the sub-occipital region, which was cleaned with betadine. By means of a vertical midline incision access was gained to the cisterna magna. Once identified a 27-gauge needle was inserted into the cisterna magna and ~0.15 mL of CSF was withdrawn into a syringe to avoid increased intracranial pressures with injection of an autologous blood volume. A total of 0.3 mL blood was extracted from the femoral artery and slowly injected into the cisterna magna. The needle was left in place for 30 s to ensure clotting in the subarachnoid space and then carefully withdrawn. Hemostasis was confirmed and the incision was closed using a stapling device. Animals were placed prone in a 20° head down position for 20 min to allow blood to congeal in the cisterns around the basilar artery.

### 2.2. Preconditioning Intervention

All animals were preconditioned at 24 h, 48 h, and 72 h after subarachnoid hemorrhage in the following manner:

For remote limb conditioning all animals were anesthetized with isoflurane, nitrous oxide 70%, and oxygen 30%. Limb ischemic preconditioning was induced with a 2 cm Critter-Cuff ^TM^ (Ramsey Medical, Inc., Ramsey, NJ, USA) applied as proximally as possible over the hind limb and inflated manually above systolic blood pressure. The inflation pressure was gauged with the use of an Oximeter Pod (AD Instruments, Colorado Springs, CO, USA) probe applied to the ipsilateral foot and monitored, to confirm a drop in O_2_ saturation, as the cuff was inflated. The limb was also observed to become dusky. The cuff was inflated for 10 min, followed by 5 min of reperfusion. During reperfusion, the limb was monitored for normalization of O_2_ saturation and limb color. At each session, three cycles of ischemia and reperfusion were completed (Figure 1). We used a 3 × 10 min conditioning paradigm. We had previously shown this to be implementable in patients with SAH [6]. Rectal temperature was monitored during the conditioning and maintained between 37.0–37.5 °C with heating lamps as necessary, and the duration of anesthesia was 25 min during each session.

Sham limb condition was conducted in the identical manner. The only difference was that the cuff was only minimally inflated around the hind limb.

Resveratrol preconditioning was carried out with intraperitoneal injections of 10 mg/kg at 24 h, 48 h, and 72 h after SAH. We used resveratrol at 10 mg/kg, which is a dose we had previously found to successfully precondition hippocampal neurons in a global cerebral ischemia model [3]. Intraperitoneal vehicle injections were delivered at the same timepoints. (Figure 1).

### 2.3. Morphometric Analysis

Four days after SAH, animals were trans-cardiacally perfused with a mixture of 50% formaldehyde, glacial acetic acid, and methanol, at 1:1:8 by volume [7]. Brains, with careful attention to the brainstem and basilar artery, were extracted and tissues were processed and embedded in paraffin. The brainstem was cut in the coronal plane into 10 µm thick slices with a motorized microtome. The proximal, mid, and distal segments of the basilar artery were used for measurement. The vertebrobasilar junction served as a point of reference. The proximal basilar artery was identified just cephalad to the vertebrobasilar junction. The mid basilar section was taken 40–60 µm cephalad to the vertebrobasilar junction. The distal section was taken 80–100 µm cephalad to the vertebrobasilar junction. Sections of interest were scanned using a PrimeHisto XE Histology Slide Scanner (Carolina Biological Supply Company, Burlington, NC, USA) at 10,000 pixels per inch with 32-bit per pixel. Measurements were made using ImageJ software and the inner cross-sectional area was measured for all three sections. All histological analysis was done blinded to group allocation.

### 2.4. Histological Analysis

Histological assessments of neuronal death were made in the hippocampus (~3.4 to 3.8 mm posterior to bregma) to assess cerebral ischemia [8]. Three sections 200 µm apart were assessed using a Nikon microscope (Nikon Microphot-SA, Nikon Corporation, Tokyo, Japan) and MCID Elite software (InterFocus Imaging Ltd., Cambridge, UK). The number of normal neurons was tabulated in CA1 hippocampus (both sides) at a magnification of 40× by an investigator blinded to the experimental conditions. The length of the CA1 region was also recorded. Total neuronal counts from both hemispheres of brain were added and normalized per mm length of CA1 region and averaged.

#### Behavioral Outcomes

Neurological and general well-being evaluations were made on day 4 and just before euthanasia. For the neurological score, a motor score (0–12; comprising spontaneous activity, symmetry of limb movement, climbing, and balance and coordination) and a sensory score (4–12; comprising proprioception plus vibrissae, visual, and tactile responses) were added together [9]. The general well-being score was adapted from a similar score in mice and consisted of observations of eyes, ears, posture, grooming behavior, spontaneous activity and presence of convulsions with a score ranging from 0–26 [10].

### 2.5. Ethics Statement

All animal studies were approved by the University of Miami, School of Medicine, Animal.

Studies Committee under guidelines and regulations consistent with the Guide for the Care and Use of Laboratory Animals, Public Health Service Policy on Humane Care and Use of Laboratory Animals, and the Animal Welfare Act and Animal Welfare Regulations.

### 2.6. Statistical Analysis

We compared true preconditioning with sham preconditioning groups and resveratrol preconditioning was compared with vehicle preconditioning using SPSS for Windows (Version 22). Chi-square was used for categorical variables and the Student *t*-Test for continuous variables. An outlier analysis was performed using Grubb’s test (GraphPad), and outlying values were excluded from analysis. One outlier was removed from the sham conditioning group and one from the vehicle group based on outlying hippocampal cell counts. Values of *p* < 0.05 were statistically significant.

## 3. Results

There were no statistically significant differences in the physiological parameters monitored between the groups (data not shown). The mortality of SAH was 9%. All study interventions were completed as planned and there were no technical difficulties with tourniquet limb conditioning. All tourniquet-conditioned animals had a reduction of O_2_ saturation in the conditioned limb. No reduction was noted in the sham-conditioned animals.

### 3.1. Behavioral Evaluations

We found no differences in the general behavioral and neurological scores between groups (Table 1).

### 3.2. Basilar Artery Analysis

In the sham cuff-treated group, the mean basilar cross-sectional area at the vertebrobasilar junction and mid and distal segments was 0.053 ± 0.014, 0.040 ± 0.017, and 0.047 ± 0.015 mm^2^, respectively. In the true cuff-treated group, the mean basilar cross-sectional area at the vertebrobasilar junction and mid and distal portion was 0.059 ± 0.016, 0.057 ± 0.011, and 0.053 ± 0.021 mm^2^, respectively. The cross-sectional area of the mid-basilar artery in the true cuff preconditioning group was significantly larger by 43% (*p* = 0.03) when compared to the sham preconditioning group. However, the diameter for the proximal and distal basilar segments were not statistically different between these two groups (Figure 2). Representative vessel diameters are shown in Figure 3. No differences in the cross-sectional area were found in the resveratrol-treated group when compared to the vehicle-treated group for any of the three basilar segments (Figure 2).

### 3.3. Histological Hippocampal Cell Counts

Mean hippocampal neurons/mm counts did not differ between the groups (Table 1, Figure 4).

## 4. Discussion

Cerebral vasoconstriction remains an important pathological finding in SAH, even though its role as the only contributor to DCI has been questioned recently [11]. In the present study we were able to show a vasoprotective effect of remote limb ischemic conditioning on basilar artery diameter; however, we were not able to demonstrate a similar effect with resveratrol. We did not find a protective effect of either conditioning paradigm on neurological and behavioral outcomes or hippocampal cell counts. In contrast to our findings, Karaoglan et al. found that three daily doses of intravenous resveratrol 10 mg/kg improved basilar artery diameters and wall thickness in a single injection rodent SAH [12]. The disparate findings may be explained by differences in the route and timing of resveratrol administration. While we injected resveratrol intraperitoneally, starting 24 h after SAH, the study by Karaoglan et al. used intravenous resveratrol, with the first dose administered immediately after SAH. It is possible that intraperitoneal and delayed resveratrol administration may not provide a sufficient conditioning stimulus after SAH. However, delaying the administration of the conditioning intervention is clinically more relevant.

There have been few studies that have directly compared different conditioning strategies. Our data suggest that remote ischemic conditioning may provide a more powerful conditioning stimulus than resveratrol. It has previously been suggested that the phenotype of the conditioning response may be determined by the nature of the conditioning stimulus [13]. In this manner, tailoring the conditioning stimulus to the targeted pathology may be a more effective strategy, i.e., using ischemia to condition for lethal ischemia rather than using another stimulus, e.g., lipopolysaccharide conditioning, which reprograms the cellular response through inflammatory mediators and may be more effective in inflammatory predominant pathologies [13].

We would like to emphasize the importance of using a clinically relevant model of experimental SAH. Table 2 summarizes preconditioning studies in animal models of SAH. Preclinical studies have included a variety of stimuli such as hyperbaric oxygen, lipopolysaccharide, and limb conditioning. Limb conditioning with a blood pressure cuff is clinically appealing, as this can be easily instituted in patients. However, the only experimental study assessing the effect of remote limb conditioning for rodent SAH used direct femoral artery occlusion with vascular clips, which is impractical in humans [14].

Preconditioning is being increasingly explored for treatment of all stroke subtypes [2]. This includes ischemic stroke, as well as intracerebral hemorrhage. In intracerebral hemorrhage, ischemic conditioning with 15 min of direct middle cerebral artery occlusion attenuated intracerebral hemorrhage-related brain edema [15]. Similar findings were noted with preconditioning with hyperbaric oxygen [16]. A proof-of-concept clinical trial of remote ischemic limb preconditioning within 24–48 h in patients with intracerebral hemorrhage is currently ongoing [17]. However, it remains uncertain whether a particular conditioning paradigm confers a therapeutic advantage. Our data suggest that there may be differences in the preconditioning potential between the various strategies.

In our study, limb preconditioning prevented vasoconstriction in the mid-basilar artery, but for reasons which are not clear, no effects on the proximal or distal segments were seen. The mid-basilar segment may be more susceptible to vasoconstriction, given that the proximal and distal segments have arterial confluences (vertebrobasilar junction) or bifurcations (top of the basilar), which may confer a more stable vascular structure. We were not able to find clinical reports showing a susceptibility of the mid-basilar artery to vasospasm in human SAH. However, Figure 5 shows a catheter angiogram of a case of mid-basilar vasospasm in a patient with recent SAH, with sparing of the proximal and distal segments.

It is also important to note that remote limb conditioning was vasoprotective despite using isoflurane anesthesia at the time of the intervention. Isoflurane is a conditioning agent and has been shown to improve microvascular function and neurological outcome in experimental SAH [18,19]. It is particularly noteworthy that despite the background of isoflurane use in both sham and true limb conditioning groups, we were able to observe an added protective effect of remote conditioning.

In our model we were not able to show a neuronal protective effect of either resveratrol or limb conditioning on hippocampal cell counts. Prior studies that found hippocampal cell loss in the single injection rodent SAH model have typically included longer survival times (7 days) than that used in our study (4 days) [20,21]. We did not a include a control without SAH. For that reason, we are uncertain if the model used leads to hippocampal cell injury by day 4 after SAH or whether the interventions are truly ineffective in preventing neuronal loss.

In most preclinical studies conditioning was started either prior to or immediately after SAH. This is not practical in the clinical setting as diagnosis, medical care, and aneurysm treatment cause delays in implementing any intervention [14,21,22].

We also believe that experimental models should include repetitive conditioning over days following SAH. In human SAH this would require repeated conditioning to cover the entire 4–14-day risk period for DCI. The conditioning response wanes within days, requiring repeated treatment [23]. It is probable, but uncertain, that the risk period for DCI is shorter in rodents than in humans, requiring fewer treatment sessions [12,24].

Our study has several limitations. We were not able to fully assess histological neuronal protection or behavioral outcomes. Cognitive function is typically affected by SAH and requires more detailed testing than we were able to do. This may be the reason why we were not able to demonstrate a beneficial effect of either intervention on behavioral outcome. In addition, we were only able to remotely pre-condition under isoflurane anesthesia. In future studies it will be important to condition animals without anesthesia to determine the true effect of limb conditioning only. We used male rats only and sex differences will need to be explored, particularly as the incidence of SAH is higher in women. The lack of a control group, without SAH and only short-term histological outcome assessment, did not allow us to fully explore any effect on hippocampal cell injury of the interventions.

In conclusion, we found beneficial vascular effects of remote limb preconditioning on basilar artery vasospasm. Our findings support further studies of remote limb preconditioning as treatment to improve outcome from SAH. We also highlight the need to include a clinically relevant conditioning model and trust that future investigators will consider adapting these important translational elements.

## Figures and Tables

**Figure 1 biomolecules-12-00568-f001:**
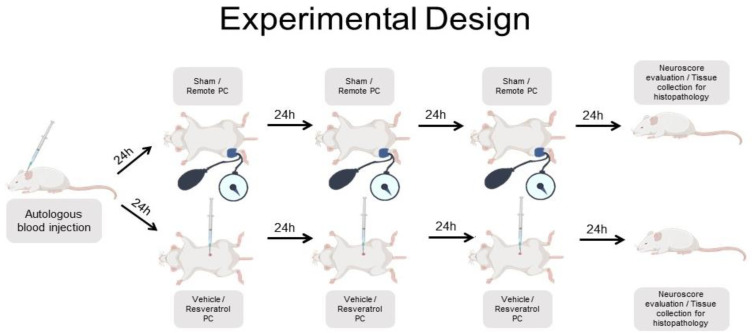
Overview of the experimental design (PC = preconditioning).

**Figure 2 biomolecules-12-00568-f002:**
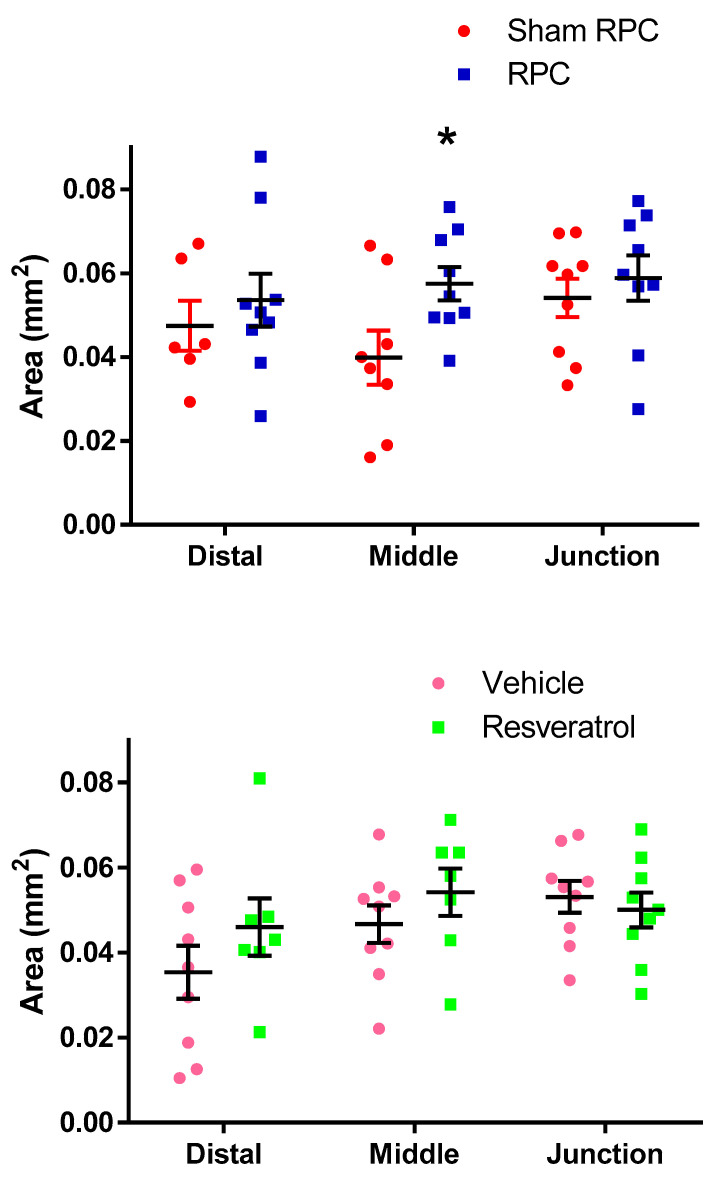
Mean cross sectional area of the basilar artery (* = *p* < 0.05).

**Figure 3 biomolecules-12-00568-f003:**
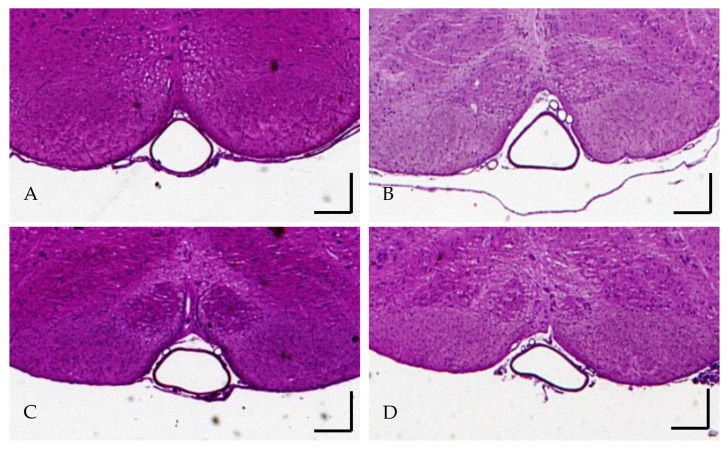
Representative examples of mid-basilar artery vascular diameters of (**A**) sham conditioning (n = 8), (**B**) true conditioning (n = 9), (**C**) vehicle (n = 9) and (**D**) resveratrol conditioning groups (n = 7) (bar length is 0.2 mm).

**Figure 4 biomolecules-12-00568-f004:**
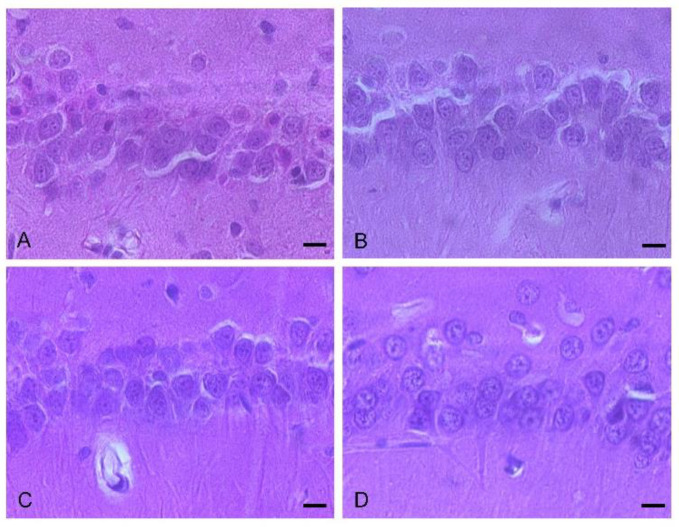
Hippocampal cell counts of (**A**) sham conditioning (n = 9), (**B**) true conditioning (n = 9), (**C**) vehicle (n = 9) and (**D**) resveratrol conditioning groups (n = 7) (bar length is 10 µm).

**Figure 5 biomolecules-12-00568-f005:**
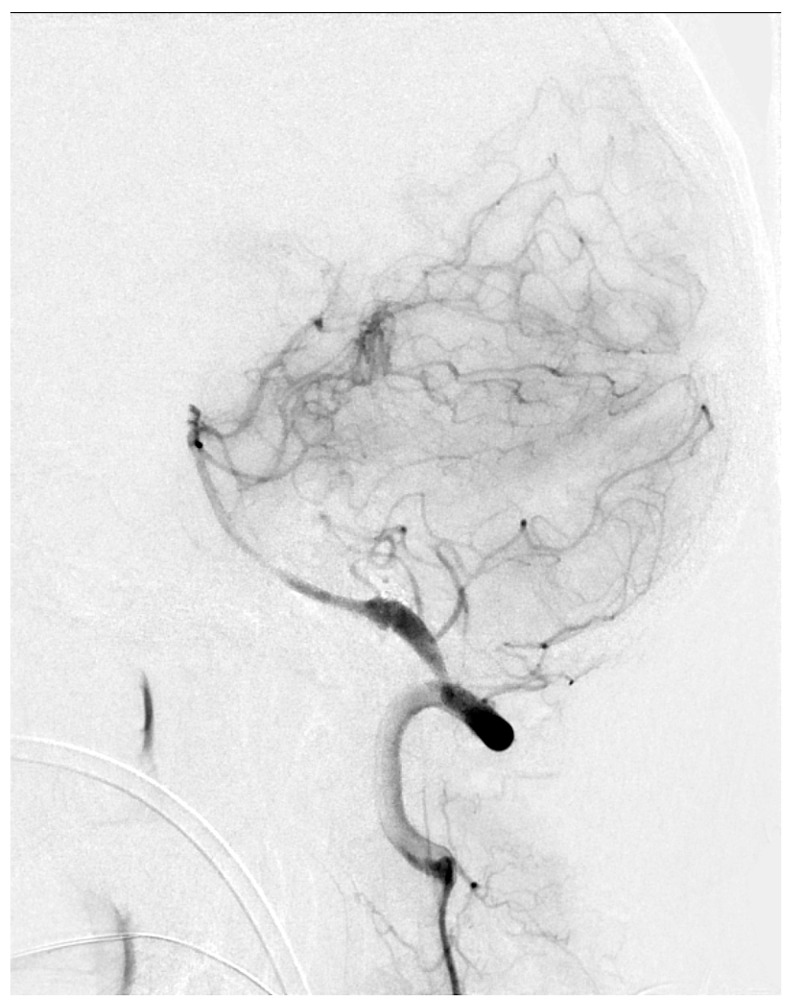
Lateral view of basilar artery catheter angiogram in a patient with recent subarachnoid hemorrhage demonstrating mid-basilar artery vasospasm.

**Table 1 biomolecules-12-00568-t001:** Behavioral scores, histological data, and basilar artery cross-sectional area.

Sham Cuff vs. True Cuff Conditioning	Sham Cuff	TRUE CUFF	*p*-Value
General score- total (median)	**2** **(n = 9)**	**2** **(n = 10)**	**0.72**
Neuro score- focal (median)	**0** **(n = 9)**	**0** **(n = 10)**	**0.34**
Hippocampal neurons/mm (mean)	**191 ± 19** **(n = 9)**	**193 ± 25** **(n = 9)**	**0.84**
**Basilar Cross-sectional area mm^2^ (mean)**			
junction	**0.053 ± 0.014** **(n = 9)**	**0.059 ± 0.016** **(n = 9)**	**0.38**
mid	**0.040 ± 0.017** **(n = 8)**	**0.057 ± 0.011** **(n = 9)**	**0.03**
Distal	**0.047 ± 0.015** **(n = 6)**	**0.053 ± 0.021** **(n = 9)**	**0.84**
**Vehicle vs.** **Resveratrol**	**Vehicle**	**Resveratrol**	***p*-Value**
General score- total (median)	**1** **(n = 6)**	**2** **(n = 8)**	**0.14**
Neuro score- focal (median)	**0** **(n = 6)**	**0** **(n = 8)**	**1.00**
Hippocampal neurons/ mm (mean)	**193 ± 28** **(n = 9)**	**181 ± 38** **(n = 9)**	**0.47**
**Basilar Cross-sectional area mm^2^ (mean)**			
junction	**0.053 ± 0.011** **(n = 9)**	**0.050 ± 0.012** **(n = 9)**	**0.61**
mid	**0.047 ± 0.013** **(n = 9)**	**0.054 ± 0.015** **(n = 7)**	**0.23**
distal	**0.035 ± 0.018** **(n = 9)**	**0.046 ± 0.018** **(n = 7)**	**0.23**

**Table 2 biomolecules-12-00568-t002:** Experimental studies of preconditioning for SAH.

Study	Animal SAH Model	Stimulus	Conditioning Start in Relation to SAH	Repetitive Conditioning	Effect
Karaoglan 2008	Rat Cisterna Magna injection	Resveratrol	One minute	yes	Reduced vasospasm
Ostrowski 2005	Rats Endovascular Perforation	Hyperbaric O_2_	60 min	no	Neuronal protection
Vellimana 2011	Mouse Endovascular perforation	Hypoxia	Prior to SAH	no	Reduced vasospasm Neurological improvement
Smithason 2013	Mouse SAH vein transection	LPS	Prior to SAH	no	Reduced vasospasm
Milner 2015	Mice Endovascular perforation	Isoflurane	60 min	no	Improved microvascular function Neurological improvement
Hu 2018	Rats Endovascular perforation	Femoral artery clip occlusion	Immediately	yes	Neurological improvement
Athiraman 2021	Mouse Endovascular perforation	Isoflurane	60 min	no	Reduced vasospasm Neurological improvement

SAH = subarachnoid hemorrhage.

## Data Availability

Data are available upon any reasonable requests in accordance with the University of Miami data transfer policies.

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
