# Peer review of "Comparing Protection of Remote Limb with Resveratrol Preconditioning following Rodent Subarachnoid Hemorrhage"

_biomolecules, 2022, doi:10.3390/biom12040568_

Round 1
Reviewer 1 Report
Cerebral vasoconstriction contributes to delayed cerebral ischemia after SAH. Intervention that prevent or treat cerebral vasoconstriction is critical for SAH treatment. In this proof of concept study, the authors found that repetitive limb conditioning with a blood pressure cuff can expand the cross-sectional area of the mid-basilar artery by 43% than sham and that remote ischemic conditioning may provide a more powerful conditioning stimulus than resveratrol. Although histological and behavioral results did not show encouraging results, repetitive limb conditioning with a blood pressure cuff showed to be beneficial to relief basilar artery vasospasm after SAH. Given the fact that SAH is life threatening and a medical emergency and there is no effective treatment, and that Repetitive limb conditioning with a blood pressure cuff is clinically relevant approach, it is well deserved to further investigate the long-term effect of this preconditioning approach in the context of SAH.
Author Response
We thank the reviewer for the comments. We agree that this is a proof of concept study and look to further investigate conditioning strategies in hemorrhagic stroke.
Reviewer 2 Report
The author compared protection of remote limb with resveratrol preconditioning in rat model of SAH. They found that remote limb effectively improved the mid-basilar artery area than sham group, while no significant differece was found between resveratrol group and vehicle group. Based on this, they came to the following conclusion beneficial vascular effects of remote limb preconditioning on SAH induced basilar artery vasoconstriction. This study has some readability, but the experimental data need to be rigorously improved.
- How many animals were assigned to each group,please add it into Table 1 and Figure legends.
- In Fig2 , please replace the existing bar chart with a scatter chart.
- The most important issue is Fig3, the magnification of these four pictures looks different, which seriously affected the data reliability. Thus, the author should provide an uniform magnification of these pictures, including a high power and a low power, and marked with scale values.
- In addiditon, a lack of a pictorial display of hippocampal neuron counts in the Result part. Please add it.
Author Response
We thank the reviewer.
Regarding the points raised, we have revised the manuscript as follows:
- We have added the number of animals for each variable analyzed in the tables and the legend as requested.
- Figure 2 is now a scatter chart.
- We have made the images in Figure 3 at a uniform magnification and added a scale bar.
- We have added representative images from the hippocampal counts in Figure 4.
Reviewer 3 Report
- The manuscript proposes to compare resveratrol preconditioning to remote limb precondition in protecting against subarachnoid hemorrhage. Besides several typos, the manuscript shows only the significant difference between the cross-sectional area of the mid-basilar artery in the limb preconditioning and sham remote conditioning groups. The punctuations should be checked: as most of the sentences are too runny and can be easily misinterpreted when read.
- Typos (to name a few):
- “R” is missing in remove in line 13
- Line 317, change “in ongoing” to “is ongoing”
- Although the authors reviewed the benefits of several precondition methods after SAH, the current study results look ordinary to me. I expect to see differences if adopting different remote limb preconditioning regimes or revealing the probable mechanism underlying remote limb precondition.
Author Response
We thank the reviewer for the feedback. We have edited the grammar and spelling to the best of our abilities.
We do agree that the negative results are disappointing and appear ordinary, and hope to investigate differences in future studies.
Round 2
Reviewer 2 Report
The author has made the corresponding modification, I have no other comments.
Author Response
Thank you.
Reviewer 3 Report
- The manuscript is hard to read with all the correction marks on it.
- Typos are occasionally found, e.g., line 258, diamters.
- The discussion is not well organized, please improve it. Some sentences are hard to understand, e.g., line 359: Our study did not a control without SAH.
Author Response
We apologize for the delay in our response. We believe we have addressed the reviewers concerns by re-organizing our discussion and having the manuscript edited by a respected member of our faculty.
We will try to upload both a marked and clean version of the manuscript if the submission site allows us to do so.
